# Targeting integrin αvβ3 by a rationally designed protein for chronic liver disease treatment

Ravi Chakra Turaga [1], Ganesh Satyanarayana[1], Malvika Sharma[1], Jenny J. Yang [2], Shiyuan Wang[3], Chunfeng Liu[3], Sun Li[2], Hua Yang[4], Hans Grossniklaus [4], Alton Brad Farris [5], Jordi Gracia-Sancho[6] & Zhi-Ren Liu[1✉]

Chronic Liver Diseases (CLD) are characterized by abnormal accumulation of collagen fibrils, neo-angiogenesis, and sinusoidal remodeling. Collagen deposition along with intrahepatic angiogenesis and sinusoidal remodeling alters sinusoid structure resulting in portal hypertension, liver failure, and other complications. Efforts were made to develop treatments for CLDs. However, the success of such treatments is limited and unpredictable. We report a strategy for CLD treatment by induction of integrin $\alpha_v\beta_3$ mediated cell apoptosis using a rationally designed protein (ProAgio). ProAgio is designed to target integrin $\alpha_v\beta_3$ at a novel site. Integrin $\alpha_v\beta_3$ is highly expressed in activated Hepatic Stellate Cells (HSC), angiogenic endothelium, and capillarized Liver Sinusoidal Endothelial Cells (LSEC). ProAgio induces apoptosis of these disease causative cells. Tests with liver fibrosis mouse models demonstrate that ProAgio reverses liver fibrosis and relieves blood flow resistance by depleting activated HSC and capillarized LSEC. Our studies demonstrate an effective approach for CLD treatment.

[1] Department of Biology, Georgia State University, Atlanta, GA 30324, USA. [2] Department of Chemistry, Georgia State University, Atlanta, USA. [3] Amoytop Biotech Inc., Xiamen, P. R. China. [4] Department Ophthalmology, Emory University, Atlanta, GA 30322, USA. [5] Department of Pathology, Emory University, Atlanta, GA 30322, USA. [6] IDIBAPS Biomedical Research Center & CIBEREHD, Barcelona, Spain. ✉email: zliu8@gsu.edu

Activation of HSC has a major role in CLD development and progression[1–4]. Owing to the critical function of HSC in collagen accumulation in the fibrotic liver, it is suggested that it would be a great benefit to specifically deplete activated HSC (αHSC) as a treatment strategy for CLD[5,6]. However, specific removal of αHSC from the fibrotic liver has not yet been achieved. Vasculature in liver sinusoids differs from that in other organs. Sinusoids are made of liver sinusoidal endothelial cells (LSEC) that possess a unique feature of fenestration and lack of basal membrane. In the fibrotic liver, inflammatory responses to liver insults activate HSC and promote LSEC dedifferentiation or capillarization, proliferation, and migration, which leads to sinusoid remodeling and intrahepatic angiogenesis[7,8]. Sinusoidal remodeling and intrahepatic angiogenesis result in dysregulated vessel structure in sinusoids[9,10]. The dysregulated vessel structure with the dysfunction of HSC often leads to resistance to blood flow in the sinusoidal space[11,12]. The consequence is liver failure and other related complications, such as portal hypertension. Interestingly, activation of HSC is tightly coupled to sinusoidal remodeling and intrahepatic angiogenesis in fibrotic liver[13–16]. In fact, sinusoidal remodeling precedes the onset of liver fibrosis and promotes the activation of HSC. In return, αHSC facilitates sinusoidal remodeling by secreting a number of molecular factors that promote LSEC growth migration and dedifferentiation[13,17], whereas the capillarized LSEC (caLSEC) maintains HSC activation[18]. In addition, ECM released by αHSC also has a role in promoting intrahepatic angiogenesis and sinusoidal remodeling[19,20]. It may be less effective to separately tackle an individual event. We envision that one agent that simultaneously depletes αHSC and caLSEC would potentially bring important advantages in the treatment of patients with CLD and associated complications.

We recently reported the development of a protein drug candidate (ProAgio) that targets integrin $\alpha_v\beta_3$ at a non-ligand-binding site. ProAgio induces apoptosis of integrin $\alpha_v\beta_3$-expressing cells by recruiting/activating caspase 8 to the cytoplasmic domain of the targeted integrin[21,22]. Considering that αHSC and caLSEC express high levels of integrin $\alpha_v\beta_3$[23–25], we reasoned whether ProAgio would be effective in inducing apoptosis of αHSC and caLSEC, and therefore represent a novel strategy for the treatment of liver fibrosis and portal hypertension.

## Results

**Integrin $\alpha_v\beta_3$-targeting protein drug ProAgio induces apoptosis of αHSC.** We first probed the expression of integrin $\alpha_v\beta_3$ in activated and inactivated human primary HSC. Indeed, integrin $\alpha_v\beta_3$ was expressed in high levels in αHSC (Fig. 1a; Supplementary Figure 7a, b). We also analyzed the integrin $\beta_3$ expression levels in different cell types of normal healthy or cirrhotic livers of murine TAA model by reverse transcription polymerase chain reaction (RT-PCR). The integrin was not expressed in HSC, LSEC, and hepatocyte in a normal healthy liver. The integrin was expressed in Kupffer cells of a normal healthy liver. However, in the fibrotic liver, the integrin was expressed in HSC, LSEC, and Kupffer cells but not in hepatocytes (Supplementary Figure 1a). We further re-analyzed the scRNA-seq data from the public domain (from Ramachandran et al.) and validated the expression of integrin αVβ3 in a multi-lineage approach in various cell types involved in the liver physiology and pathology (GSE136103)[26]. Our analyses indicated that, although integrin αv was expressed in several cell lineages of both normal and fibrotic liver, integrin β3 was not expressed or expressed in low levels in most cells in the liver (Supplementary Figure 1b, c). We then tested whether ProAgio added to culture media would induce apoptosis of αHSC. Apoptosis was induced in the αHSC, whereas no apoptosis

was observed in HSC without activation (Fig. 1c and Supplementary Figure 2b). To confirm that apoptosis induction was owing to integrin $\alpha_v\beta_3$ targeting by ProAgio, we used the immortalized and activated human HSC line LX-2 cells. Integrin $\alpha_v\beta_3$ was detected in LX-2 cells (Supplementary Figure 2a). ProAgio-induced LX-2 cell apoptosis (Fig. 1d and Supplementary Figure 2c), and importantly, when integrin $\beta_3$ was knocked down (Supplementary Figure 2d; Supplementary Figure 7e), ProAgio failed to induce apoptosis (Fig. 1e). As previous data demonstrated that ProAgio-induced apoptosis in endothelial cells by recruiting caspase 8 to the cytoplasmic domain of integrin $\beta_3$[21], we questioned here whether apoptosis of αHSC was induced via a similar mechanism. We observed co-precipitation of caspase 8 with integrin $\beta_3$ in LX-2 cells upon ProAgio treatment, whereas this was not observed in cells treated with vehicle (Fig. 1f; Supplementary Figure 7b). Furthermore, ProAgio treatment abolished FAK phosphorylation accompanied with activation of caspase 8 and caspase-3 in cells (Supplementary Figure 2e; Supplementary Figure 7f and Supplementary Figure 2g; Supplementary Figure 7h). To test specific effects of ProAgio via integrin $\alpha_v\beta_3$, we carried out co-IP of ProAgio with different integrins (β1, β3, β5, β6) in extracts of LX-2 cells treated with ProAgio. Clearly, ProAgio only co-IPed with integrin β3 (Supplementary Figure 2f; Supplementary Figure 7g). We conclude that ProAgio induces apoptosis of αHSC by recruitment/activation of caspase 8 at the cytoplasmic domain of integrin $\beta_3$.

**ProAgio reverses liver fibrosis.** To test whether ProAgio exerts its activity in vivo, we used an established liver fibrosis mouse model where Balb/C mice received TAA and 10% ethanol in drinking water for 12 weeks[27]. At the end of fibrosis induction, the animals were treated with ProAgio or vehicle (Fig. 2a). ProAgio treatment led to an increase in body weight and a decrease in liver weight compared with those of the vehicle-treated group (Fig. 2b, c). The liver surface of ProAgio-treated animals exhibited fewer fibrotic features compared with that of the vehicle-treated group (Supplementary Figure 2h). Recovery of fibrotic liver damage by ProAgio was also demonstrated by the reduction of apoptotic bodies in the liver of the treated animals (Fig. 2d). Examination of serum markers of liver damage showed high levels of AST, ALT, and ALP in fibrotic animals receiving the vehicle in comparison with control, whereas ProAgio significantly improved them to the levels close to those of non-fibrotic mice (Supplementary Figure 2i). Analysis of hepatic fibrosis confirmed that animals that received TAA and were treated with vehicle exhibited significant collagen deposition within the liver parenchyma, whereas Sirius red stain of liver sections demonstrated lesser and thinner collagen accumulation in the livers of ProAgio-treated animals, especially the dense collagen networks disappeared (Fig. 2d, e). Hydroxyproline level in liver tissue accurately reflects CLD progression and fibril accumulation. We measured the hydroxyproline levels in the liver tissue of ProAgio-treated animals. Evidently, ProAgio reduced hydroxyproline levels in the fibrotic liver (Fig. 2f). ProAgio also effectively reversed TAA-induced liver fibrosis under the condition that the TAA was continually dosing along with the ProAgio treatment (Supplementary Figure 3a–c). Under continuous TAA dosing, ProAgio still reduced α-SMA levels in the fibrotic liver (Supplementary Figure 3d–e: Supplementary Figure 7i). These experiments suggest that ProAgio effectively reverses liver fibrosis. To verify the effectiveness of ProAgio on liver fibrosis reversal, a second model of liver fibrosis was used. After fibrosis induction by chronic $CCl_4$ administration, animals were treated by ProAgio (Supplementary Figure 4a). Similar to our observation with the model of TAA-alcohol-induced liver fibrosis, ProAgio reversed

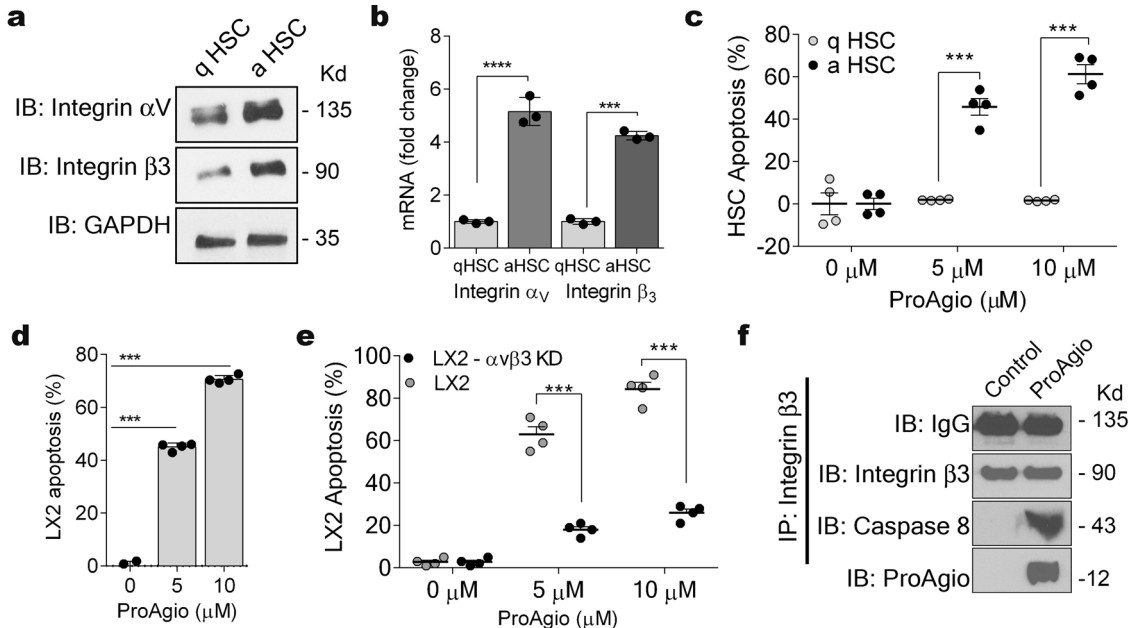

**Fig. 1 ProAgio induces apoptosis of activated HSC. a** Expression of integrin αv and β3 in activated (an HSC) and inactivated (q HSC) human primary HSC cells were analyzed by immunoblots using anti-α_v (IB: integrin αv) and anti-β_3 (IB: integrin β3) antibodies. Immunoblot of GAPDH (IB: GAPDH) is a loading control. **b** mRNA levels of integrin αv and integrin β3 in activated (an HSC) and inactivated (q HSC) human primary HSC cells were analyzed by RT-PCR. The integrin mRNAs are presented as fold changes by defining the mRNA levels of inactivated HSC as 1. **c–e** Apoptosis of human primary HSC cells (**c**), LX-2 cells (**d**, **e**) under treatment of indicated concentrations of ProAgio was measured by apoptosis kit. **f** Co-immunoprecipitations of caspase 8 with integrin β_3 (IP:integrin β_3) were analyzed by immunoblots (IB:caspase 8). LX-2 cells were treated with ProAgio 5 h prior to the preparation of extracts. Immunoblot of integrin β_3 (IB:integrin β_3) and IgG (IB:IgG) indicates the amount of β_3 and amounts of IgG, which were precipitated down in the co-IPs. **c–e** Apoptosis was presented as apoptosis (%) by defining buffer treated cells as 0%. **a–c** Primary HSC cells were activated by culturing 7 days in presence of 5 ng/ml TGF-β (an HSC). Freshly plated cells were regarded as inactivated HSC cells (q HSC). Error bars in **b–e** are standard deviations of five independent experiments. Statistical significance was calculated by two-tailed unpaired Student's t test, ***P < 0.001.

the CCl$_4$-induced liver fibrosis and recovered consequential liver damages by fibrosis induction (Supplementary Figure 4b–g).

**ProAgio resorbs collagen and reduces collagen cross-links in fibrotic liver.** In the fibrotic liver, αHSC not only produces most of the collagen, but it is also the main player in providing protection to the collagen from degradation by altering tissue levels of metalloproteases (MMPs) and tissue inhibitor of metalloproteinases (TIMPs)[19,28,29]. To test whether ProAgio treatment would alter MMPs and TIMPs levels, we examined the expression of TIMP1, TIMP2, MMP2, and MMP9 in liver extracts by ELISA. Both TIMP1 and TIMP2 levels were reduced in livers of ProAgio-treated animals compared with those of the vehicle-treated group. Similarly, the levels of MMP2 and MMP9 were reduced (Fig. 3a–d). Collagenase activity of the liver extracts exhibited a greater twofold increase in ProAgio-treated animals compared with that of vehicle-treated groups (Fig. 3e). Analyses of α-SMA mRNA and protein levels in the liver tissue by immunoblots and RT-PCR demonstrated that ProAgio reduced α-SMA-positive cells in the fibrotic liver (Fig. 3f, g; Supplementary Figure 7c). To test whether ProAgio treatment indeed induced apoptosis of αHSC in fibrotic livers, sections from the harvested livers were immunofluorescence (IF) stained via a myofibroblast marker α-SMA. IF co-stain of α-SMA and activated caspase-3 suggested that ProAgio treatment led to apoptosis of αHSC in fibrotic livers (Fig. 3h). Analyses of collagen crosslink, a hallmark of fibrosis progression to cirrhosis, in the Sirius red stains using polarized light microscope[30] revealed that ProAgio treatment led to substantially lesser collagen cross-links compared with that of the vehicle-treated group (Fig. 4a, b). In consistent with the observed reduction in collagen crosslinking, ProAgio led to reduced levels

of lysyl oxidase (LOX), the key enzyme in collagen crosslinking[31], both in protein (Fig. 4c; Supplementary Figure 7d) and mRNA (Fig. 4d) and the enzyme activity (Fig. 4e) in the liver tissue extracts and serum of the treated animals compared with the vehicle-treated animals.

**ProAgio reduces intrahepatic angiogenesis, reverses sinusoidal remodeling, and reduces portal hypertension.** One main complication encountered by CLD patients is portal hypertension, a leading cause of death and liver transplantation, owing to excessive deposition of collagen fibrils, intrahepatic angiogenesis, and sinusoidal remodeling in fibrotic liver[13,32–34]. We reasoned that ProAgio might have an effect on intrahepatic angiogenesis and sinusoidal remodeling. Intrahepatic microvascular dysfunction in the fibrotic liver is characterized by an increase in angiogenic caLSEC and a decrease in differentiated LSEC, as well as disruption of normal sinusoidal vessel networks[14]. Evidently, the TAA/alcohol fibrosis induction decreased differentiated LSEC levels, whereas ProAgio treatment led to a recovery of differentiated LSEC levels and restoration of the sinusoidal vessel network similar to that of the non-fibrotic liver (Fig. 5a, b). ProAgio treatment also led to a decrease in angiogenic caLSEC in the fibrotic liver to levels similar to that of the non-fibrotic liver as demonstrated by IHC staining of CD31 (Fig. 5c, d), an angiogenesis marker that expresses on caLSEC but not on differentiated LSEC[18,35]. Reduction in CD31 IHC staining suggested a possibility that ProAgio treatment might induce apoptosis of caLSEC in fibrotic livers. IF co-stains of CD31 with activated caspase-3 suggested that ProAgio treatment indeed led to apoptosis of caLSEC in fibrotic lives (Fig. 5e). To further understand the effects of ProAgio on sinusoids in the fibrotic liver, we

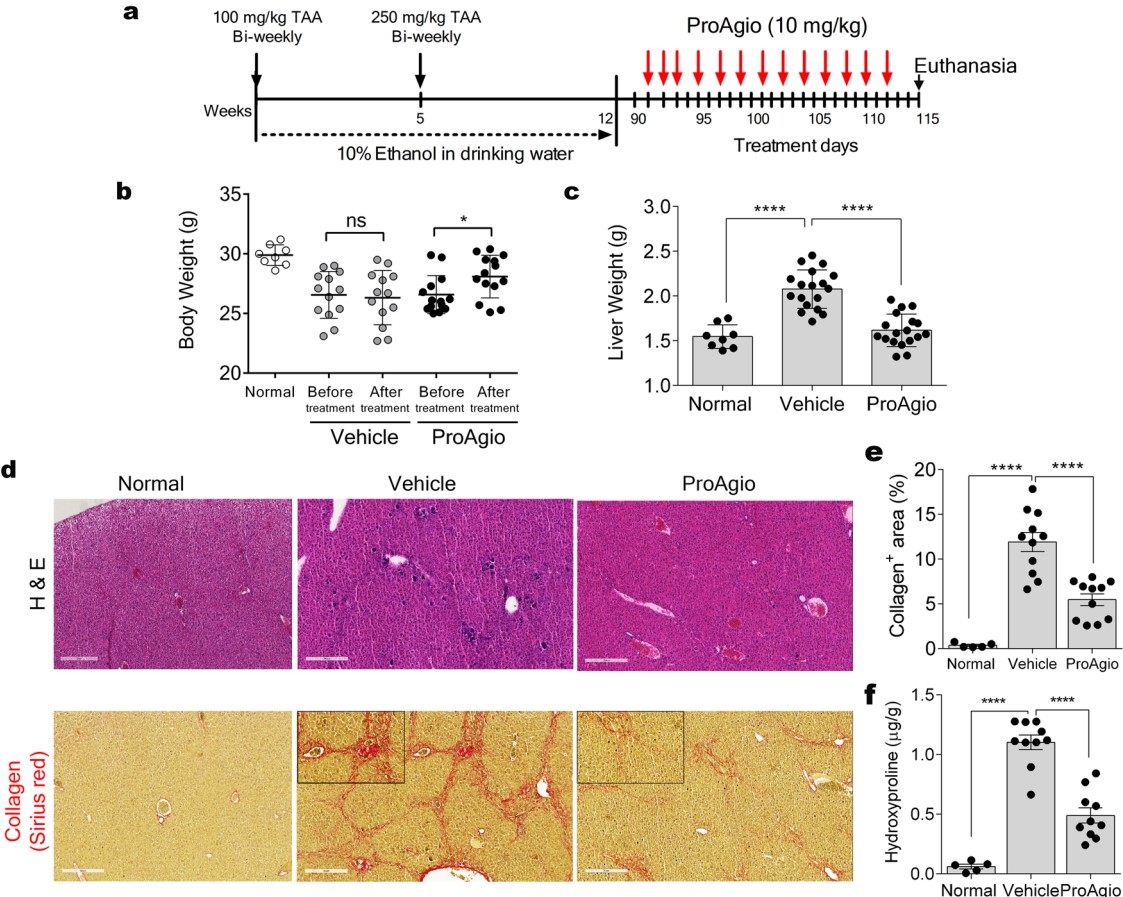

**Fig. 2 ProAgio reverses liver fibrosis. a** Schematic illustration of the schedule of TAA/alcohol liver fibrosis induction and subsequent ProAgio treatments (red arrows). **b** Body and **c** liver weight of the animals at the endpoint of experiments (**b**) or at end of fibrosis induction (before treatment, Before) or after ProAgio treatment (After) in **c**. **d** Representative images of H&E (Upper) Sirius red (Bottom) staining of sections of liver tissues from mice treated with indicated agents. Scale bars show 300 μm. **e** Quantitation of collagen levels in Sirius red staining using ImageJ software. Four randomly selected tissue sections per animal and three randomly selected view fields in each section were quantified. The quantity of collagen levels is presented as % of the total area. **f** Hydroxyproline levels in liver extracts of mice treated with indicated agents. The data in **e** and **f** were quantified from measurements of 10 mice. Error bars in **e** and **f** are standard deviations of measurements of 10 mice. Normal in **b–e** and **f** are the mice without fibrosis induction and treatment. *$P < 0.05$; ****$P < 0.0001$; ns. denotes not significant.

analyzed tissue sections by SEM imaging. Clearly, the percent of porosity areas on sinusoids decreased upon fibrosis induction, a typical characteristic of loss of sinusoidal fenestrations (Supplementary Figure 5a). ProAgio treatment led to a recovery of percent of porosity on sinusoids almost to the level of the non-fibrotic liver (Fig. 6a–c). A long-noted unique feature of sinusoidal remodeling during CLD progression is the filling of the space of Disse by collagen accumulation, which is a major contributor to intrahepatic pressure[36,37]. It is evident from the SEM-imaging analyses that ProAgio treatment led to a recovery of the space of Disse (Fig. 6d, e, arrows in Fig. 6d).

Measurement of hepatic portal vein blood flow by Doppler ultrasound imaging[38,39] demonstrated that fibrosis induction resulted in a significant reduction in hepatic in-flow, thus confirming an increment in the hepatic vascular resistance, whereas ProAgio treatment recovered liver perfusion almost to the level of the non-fibrotic liver (Fig. 6f and Fig. Supplementary Figure 5b, c). Considering that one consequence of blood flow resistance is hypoxia[40,41] we carried out analyses of Hif-1α in liver sections from the different experimental groups. Importantly, fibrotic animals treated with vehicle exhibited increased Hif-1α levels, whereas ProAgio treatment reduced Hif-1α to a level similar to that of the non-fibrotic liver (Supplementary Figure 5d, e). Altogether suggest

that ProAgio was indeed able to ameliorate intrahepatic vascular resistance, and presumably portal hypertension.

**ProAgio specifically affects αHSC and caLSEC in fibrotic liver.** We finally aimed to assess whether ProAgio induces apoptosis in other cell types of the liver. To answer this question, healthy mice were treated by ProAgio. We then carried out histological analyses and assessed the status of hepatocytes, Kupffer cells, and LSECs in tissue sections from the vehicle and ProAgio-treated animals. Clearly, ProAgio treatment did not lead to any notable anatomic changes judged by histological analyses of the liver sections from the treated mice (Supplementary Figure 6a). ProAgio also did not lead to any change in levels of LSEC and Kupffer cells (Supplementary Figure 6b, c). To further test the effects of ProAgio on other cell types in the liver, primary human hepatocyte cells, and Kupffer cells were obtained from commercial sources. Primary human hepatocyte cells do not express integrin $\alpha_v\beta_3$ (Supplementary Figure 6d; Supplementary Figure 7j). The integrin was also not expressed in inactivated LSEC. These primary cells were treated by ProAgio and no induction of apoptosis was observed (Supplementary Figure 6d; Supplementary Figure 7e, f, j, also see Fig. 1c). To test whether ProAgio induces apoptosis in the non-fibrotic liver, we performed

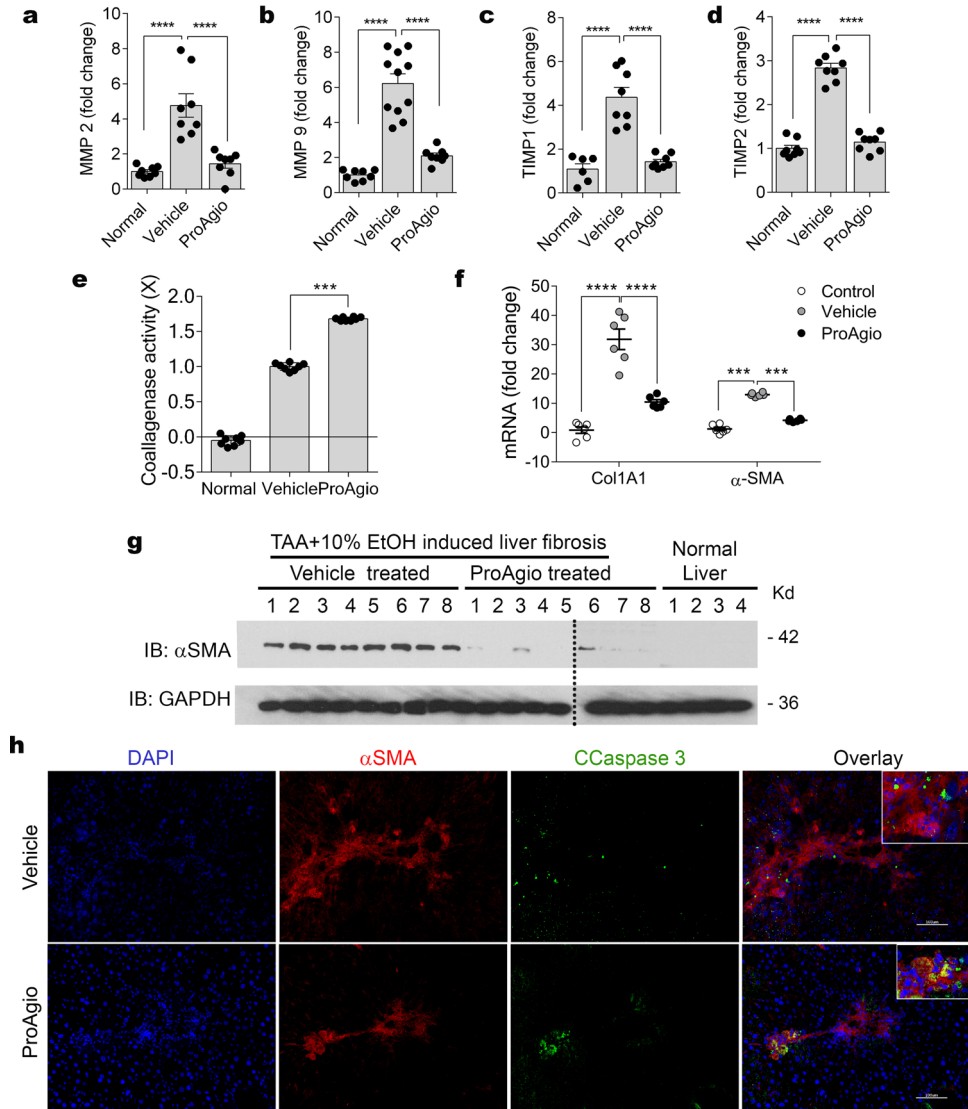

**Fig. 3 ProAgio changes collagenase activity and decreases collagen cross-links in fibrotic liver.** Levels of MMP2 (**a**), MMP9 (**b**), TIMP2 (**c**), and TIMP1 (**d**) in extracts of liver tissues from mice treated with indicated agents were measured by ELISA, and are presented as fold changes compared with that of non-fibrotic mice. **e** Collagenase activity of extracts prepared from livers of mice treated by the indicated agents was measured using Colorimetric Collagenase Activity Assay Kit. Collagenase activity is presented as fold changes using the extracts from livers of normal healthy mice as reference. **f** RT-PCR analyses of α-SMA and collagen 1 (Col1A1) mRNA levels in liver extracts of animals under indicated treatment. The mRNA levels are presented as fold changes comparing to the control (non-fibrotic and untreated animals) as reference. **g** Immunoblot analyses of α-SMA (IB: α-SMA) levels in liver extracts of the individual mice (numbered on top of picture) under indicated treatment. Immunoblot of GAPDH (IB: GAPDH) is a loading control. **h** Representative images of immunofluorescence (IF) staining of α-SMA (Red) and cleaved caspase-3 (green) and the overlay of co-stains (Yellow) of liver sections from mice treated with indicated agents. Scale bars show 100 μm. Error bars in **a**–**e** and **f** are standard deviations of measurements of 10 mice. Normal in **a**–**e** and **g** and control in **f** are the mice without fibrosis induction and treatment. ***$P < 0.001$, ****$P < 0.0001$.

immunohistochemistry (IHC) staining of cleaved caspase-3 in sections of the liver from the vehicle or ProAgio-treated normal mice (without fibrosis induction). Evidently, ProAgio did not lead to significant staining of cleaved caspase-3 (Supplementary Figure 6g), indicating that ProAgio does not induce apoptosis in a normal healthy liver. We conclude that ProAgio specifically induces apoptosis of disease-associated αHSC and caLSEC in fibrotic liver.

## Discussion

Liver fibrosis/cirrhosis is a common consequence of most chronic liver diseases. As the disease progresses, liver fibrosis/cirrhosis patients will very frequently encounter portal hypertension with symptoms such as ascites build-up. Despite intensive research in the development of therapeutics, no effective treatment agent is

available for liver fibrosis/cirrhosis patients. There is also no effective treatment for ascites build-up resulted from hepatic portal hypertension[11]. We report here a protein drug candidate, "ProAgio", which specifically induces apoptosis of αHSC and caLSEC by targeting integrin $\alpha_v\beta_3$ at a novel site. By depletion of αHSC and caLSEC, ProAgio allows collagen resorption (including reduces collagen crosslinking), reverses liver sinusoidal remodeling, and abrogates intrahepatic angiogenesis in the fibrotic liver, which consequentially reverses liver damages and relieves the hepatic portal hypertension resulted from fibrosis/cirrhosis. An important advantage is that ProAgio simultaneously and specifically depletes two disease causative cell types, αHSC and caLSEC, in liver fibrosis progression (Fig. 6g). The dual actions of ProAgio in the fibrotic liver certainly bring a very important

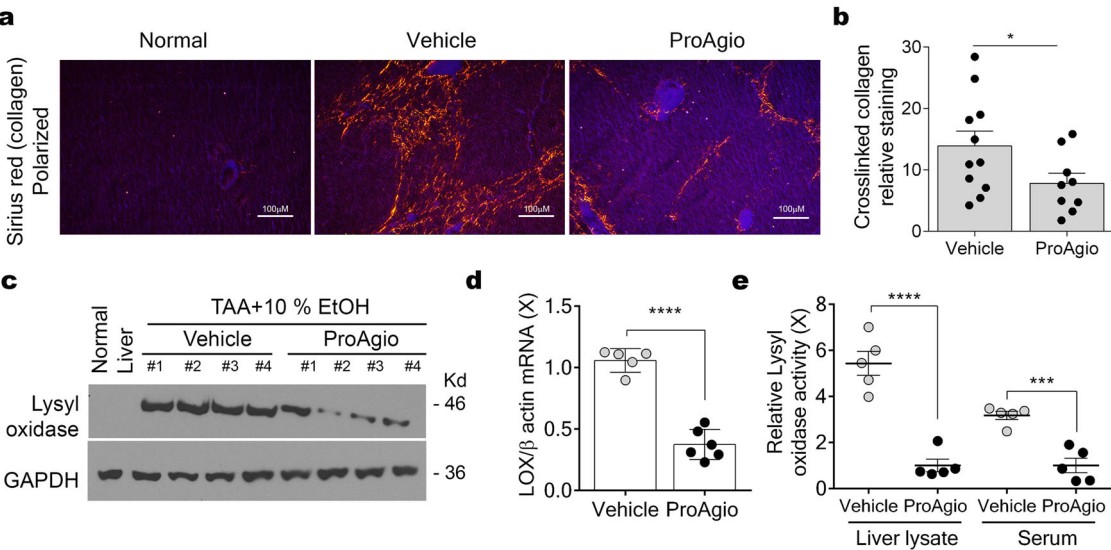

**Fig. 4 ProAgio reduces collagen crosslinking in fibrotic liver. a** Representative images of Sirius red staining of liver tissue sections from mice treated with indicated agents viewed under polarized light, indicating cross-linked collagen. Scale bars show 100 μm. **b** Quantitation of collagen levels in Sirius red stain under polarized light using ImageJ software. The quantity of cross-linked collagen levels is presented as relative staining of sections from non-fibrotic mice as reference. Quantifications of Sirius red stains under polarized light were calculated from measurements of 10 mice. Four randomly selected tissue sections per animal and three randomly selected view fields in each section were quantified. **c, d** Immunoblot (**c**, Lysyl oxidase) and RT-PCR **d** analyses of LOX protein and mRNA levels in liver extracts of four and five mice treated by indicated agents. Immunoblot of GAPDH (GAPDH) in **c** is a loading control. The mRNA levels are presented as a relative ratio compares to mRNA levels of β-actin. **e** LOX activity in liver lysate and serum of five mice treated by indicated agents was measured by LOX activity kit. The LOX activity is presented as relative LOX activity of vehicle with respect to ProAgio by defining the mean value of ProAgio-treated group as 1. Error bars in **b** are standard deviations of measurements of 10 mice. Normal in **a** is the mice without fibrosis induction and treatment. *$P < 0.05$; ***$P < 0.001$, ****$P < 0.0001$.

advantage of breaking down the vicious cycle of angiogenesis-fibrogenesis in the treatment of CLD.

Expression of integrin $\alpha_v\beta_3$ is elevated upon HSC activation[23–25]. Although function(s) of this upregulation in HSC activation is not fully understood, it was speculated that targeting this integrin using the integrin ligand mimics cilengitide might be a good approach to target αHSC for liver fibrosis. Unfortunately, although cilengitide exhibited effects in inhibiting the proliferation of αHSC in vitro, experiments with mouse models indicated that cilengitide exuberated the disease. Analyses showed that cilengitide did not induce apoptosis of αHSC in the fibrotic liver[23,24]. On the contrary, ProAgio induces apoptosis of αHSC in the fibrotic liver by targeting integrin $\alpha_v\beta_3$, which clearly is an advantage over the agents that target the ligand-binding site. Accompanying liver fibrosis progression, differentiation, or capillarization of LSEC leads to alteration of liver blood vessels. In the fibrotic liver, activation of HSC and capillarization of LSEC are tightly coupled. This tightly coupled regulation makes one event affect the other. Therefore, it may be less effective by targeting one event individually. Apparently, simultaneously targeted depletion of both αHSC and caLSEC would be beneficial for the treatment of liver fibrosis/cirrhosis and associated complications.

We have shown that ProAgio induces apoptosis of CD31-positive angiogenic endothelial cells or caLSEC apoptosis in the fibrotic liver, which subsequently leads to an increase in SE-1-positive LSEC in the treated liver. It is debatable whether CD31 is expressed in LSEC in normal healthy liver[35,42–44]. One possible reason for different observations is the use of different antibodies and detection methods in different studies. It is intriguing how ProAgio promotes an increase in SE-1-positive LSEC. One possible mechanism is that, as ProAgio depletes integrin $\alpha_v\beta_3$-positive caLSEC, the differentiated LSEC replenish themselves. As differentiated LSEC is integrin $\alpha_v\beta_3$ negative, ProAgio does not

target the differentiated LSEC. An alternative explanation is that ProAgio may trigger a reversal of capillarization of caLSEC via activation of integrin $\alpha_v\beta_3$ signaling. It will be interesting to delineate the molecular mechanism by which ProAgio promotes the increasing of differentiated LSEC in treated fibrotic liver. As maintenance of LSEC differentiation is an important factor for liver regeneration, it will be very interesting to examine whether ProAgio facilitates liver regeneration from damages caused by fibrosis/cirrhosis.

## Methods

**Reagents, antibodies, and cells**. All reagents, antibodies, commercial kits, and cell lines used in this study are listed in Supplementary Table 1 in the online supplementary information.

**Isolation of HSCs, LSECs, hepatocytes, and Kupffer cells**. HSCs, LSECs, hepatocytes, and Kupffer cells were isolated according to previously established protocols[45–47].

*Primary human hepatic stellate cells and activation.* Primary human hepatic stellate cells were purchased from Sciencell Online Catalog #5300 and Creative bioarray #CSC-C1496. The cells were cultured as per the vendor's instructions. HSC's were activated by culturing them for 1 week in presence of TGFβ 5 ng/ml.

*Primary human hepatocyte cells.* Primary human hepatocyte cells and the necessary media components were purchased from Lonza (HUCPI) and were thawed in thawing media (MCHT50), plated using plating media (MP100), and maintained in maintenance media (CC-3198).

*LX-2 cell line.* LX-2 human hepatic stellate cell line was purchased from EMD Millipore (SCC064) and was thawed in Dulbecco's Modified Eagle Medium (DMEM) containing 10% FBS, 1× Pen/Strep, and 1× Glutamine. Once established, they were cultured in DMEM containing 2% FBS along with the above-mentioned components.

*Primary human Kupffer cells.* Human Kupffer cells were purchased from ThermoFisher Scientific. Kupffer thawing/plating medium from the vendor was used,

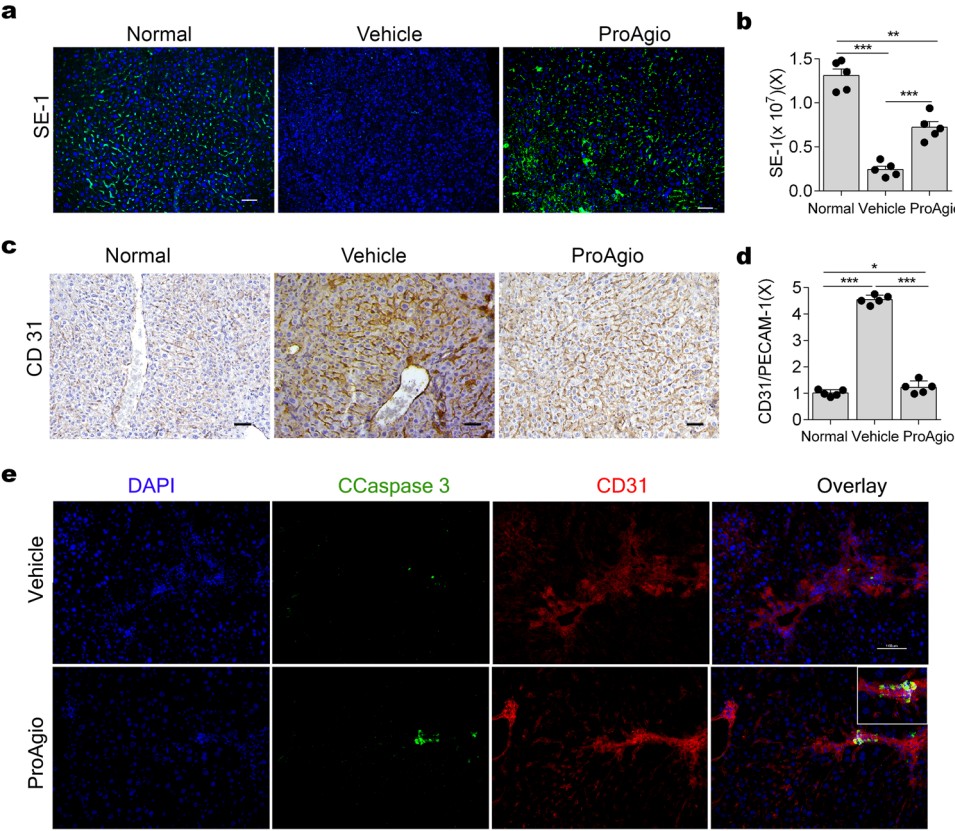

**Fig. 5 ProAgio decreases CD31-positive cells and increases differentiated LSEC in fibrotic liver. a** Representative images of IF staining of SE-1, an antibody recognizing a differentiated LSEC marker, and **b** quantitation of SE-1 IF staining of liver tissue sections from mice treated with indicated agents. The quantity of SE-1 IF is presented as a fold change in SE-1 stain compared with that of non-fibrotic mice (normal). **c** Representative images of IHC stain of CD31 and **d** quantitation of CD31 IHC stain of liver tissue sections from mice treated with indicated agents. The quantity of CD31 IHC is presented as a fold change in CD31 stain compared with that of non-fibrotic mice (normal). **e** Representative images of IF staining of CD31 (Red) and cleaved caspase-3 (green) and the overlay of co-stains (Yellow) of liver sections from mice treated with indicated agents. Scale bars in **a**, **c**, and **e** show 100 μm. Error bars in **b** and **d** are standard deviations of measurements of 10 mice. Normal in **a–d** are the mice without fibrosis induction and treatment. *$P < 0.05$; **$P < 0.01$, ***$P < 0.001$.

Kupffer cells were activated by adding lipopolysaccharide (1 μg/mL) to the culture medium for 2–24 h (depending on experimental design) prior to the experiment in either Kupffer cell monoculture or co-culture to mimic liver inflammation.

**Liver fibrosis/cirrhosis induction and treatments**. All animal experiments were carried out in accordance with the guidelines and were approved by IACUC of Georgia State University. At the end of the treatments, animals were killed. Organs and blood samples were collected for subsequent analyses. Statistical analyses were done in comparison with the control group.

*TAA-induced liver fibrosis.* Seven to eight weeks old Balb/c mice were treated twice per week via intraperitoneal (i.p.) injection of thioacetamide (TAA) at 100 mg/kg body weight (BW) for the first 5 weeks followed by 250 mg/kg up to 12 weeks along with 10% ethanol in drinking water. The TAA injection and 10% ethanol were withdrawn during buffer or ProAgio treatment (Fig. 2) and only 10% ethanol was withdrawn (for experiments in Figure S2). Healthy controls were given only an adequate saline solution by i.p. injection. Mice were killed 2 days after the last treatment.

*$CCl_4$-induced liver fibrosis.* Carbon tetrachloride was administered by a dorsal subcutaneous (SC) injection (1:1 dissolved in olive oil; 2 ml/kg) twice weekly. Administration of $CCl_4$ was stopped during ProAgio treatment.

**Collagenase assay**. Colorimetric Collagenase Activity Assay Kit was used to determine collagenase activity in liver tissue lysates, and the assay was performed by following the vendor's instruction with 1 mg of protein in the tissue lysate.

**Tissue section staining and image analyses**

*Sirius Red.* Sirius Red staining was carried out using NovaUltraTM Sirius Red Stain Kit from IHC WORLD by following the instruction of the vendor.

*H&E, IHC, and IF.* H&E, IHC, and IF staining procedures were similar to those of the previous reports[21]. Images were captured at ×20 lens aperture and scale bars indicate 50 μM in length.

*Image analyses.* H&E, IHC, and Sirius Red imaging were performed on Nano-zoomer 2.0 HT (Hamamatsu) whole-slide scanner to generate high-resolution digital images. IF imaging was captured by confocal or fluorescence microscope. Quantifications were carried out by FIJI ImageJ software.

**RNA extraction and RT-PCR**. RNA was extracted using TRIZOL reagent following the manufacturer's protocol. In all, 2 μg RNA was converted to cDNA using ThermoFisher Maxima First-strand cDNA kit. PCR was performed using primers for Integrin β3: FP-CTCCCCTCCGCAGGAAAA; RP-TCATCTGGCCGTAGTCG AAGG and for beta-actin: FP-AGGGAAATCGTGCGTGACAT; RP-GGGTGT AAAACGCAGCTCAG. Gel electrophoresis was performed to estimate gene expression.

**Hydroxyproline assay**. Hydroxyproline was measured using the kit. In brief 200 mg of tissue was homogenized in 500 mL of water. In all, 500 mL of concentrated hydrochloric acid (~12 M) was added and hydrolyzed at 120 °C for 3 h. The solution was mixed and centrifuged at $10,000 \times g$ for 3 mins. In all, 50 μL of supernatant was transferred to a 96-well plate. The plate was placed in a 60 °C oven to dry the samples. In all, 100 μL of the chloramine T/oxidation buffer mixture was added to each well. In all, 100 μL of the DMAB Reagent was added to each well after 5 mins incubation at room temperature. The plate was further incubated for 90 mins at 60 °C. Absorbance was then measured at 560 nm.

**Liver tissue SEM analyses**. The mouse liver samples were received in 2.5% glutaraldehyde fixative in 0.1 M cacodylate overnight after which they were subsampled and placed into 0.1 M cacodylate buffer. The tissue was post-fixed in 1% osmium tetroxide in 0.1 M cacodylate for 1 hour then rinsed in deionized water.

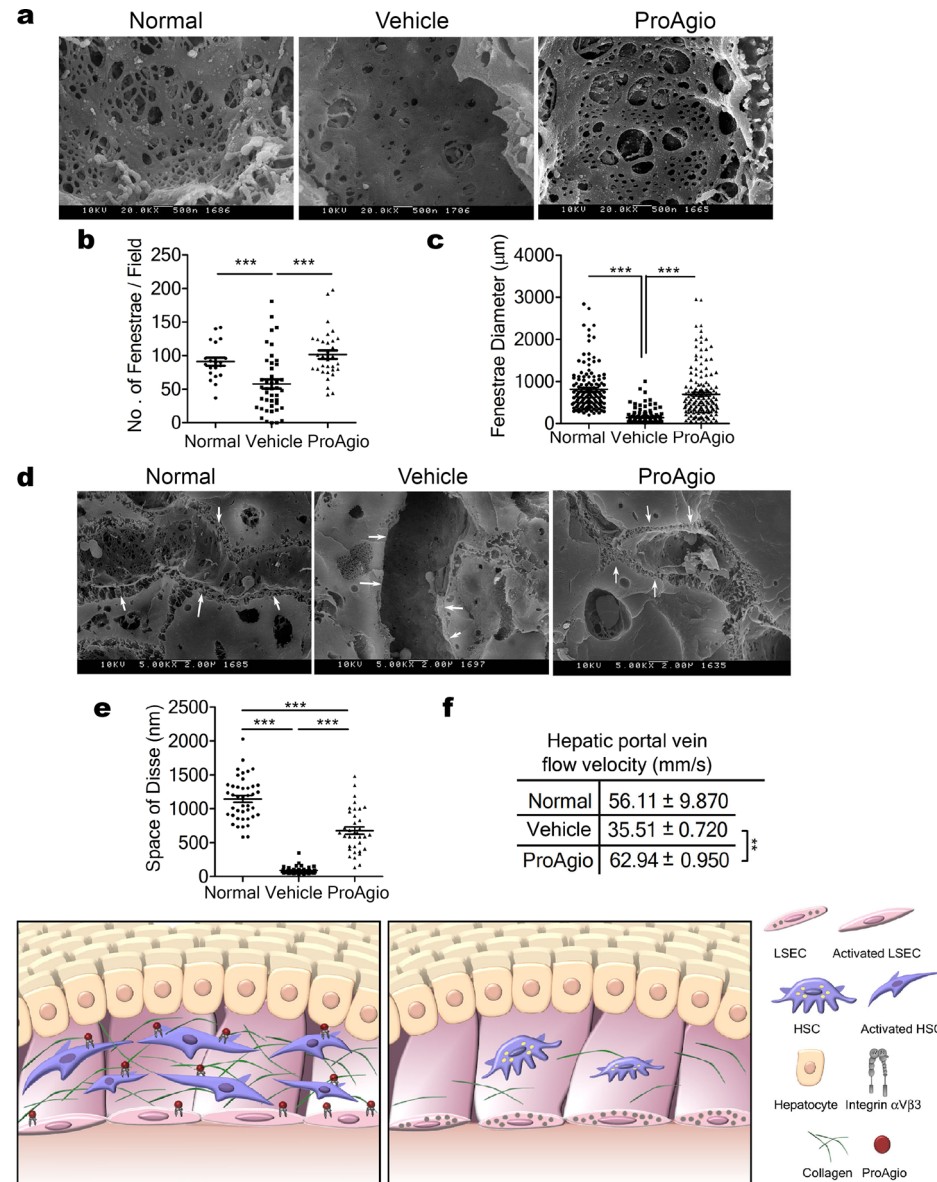

**Fig. 6 ProAgio reduces intrahepatic angiogenesis and reverses liver sinusoidal remodeling. a, d** Representative SEM images of sections from mice treated with indicated agents. Scale bars are shown 500 nm in (**a**), 2 µm in **d**. Arrows in **d** indicate the space of Disse in SEM images. Quantitation of number (**b**) and size (**c**) of fenestrations of liver sinusoids and space of Disse (**e**) in the sections of mice treated with indicated agents were measured by manually counting/measuring the number and the diameters of fenestration in the SEM images. Fenestration size is the average diameter (in µ Meter) of fenestrate from the SEM image analyses. The space of Disse (in nanometer) was manually measured in the SEM images using ImageJ software. **f** The velocity of portal vein blood flow is measured by Doppler ultrasound imaging. The flow velocity is presented as mm per second. Quantifications were calculated from measurements of 10 mice in **b**, **c**, and **e**, and five mice in **f**. Normal in **a**–**f** are the mice without fibrosis induction and treatment. **g** Cartoon illustration of the drug actions of ProAgio in reverse chronic liver disease. **P < 0.01, ***P < 0.001.

The tissue was dehydrated through an ethanol series then individual pieces were placed into a little Parafilm™ packet filled with 100% dry ethanol. Each packet was immersed in liquid nitrogen. Although immersed in the liquid nitrogen, each packet was stuck with a precooled razor blade so as to fracture the tissue sample. Each of the fractured pieces was collected and placed into a labeled vial filled with 100% dry ethanol. The dehydrated tissue samples were placed into labeled microporous specimen capsules and loaded into the sample boat of a chilled Polaron E3000 critical point drying unit. The unit was sealed and filled with liquid $CO_2$ under pressure. The $CO_2$ was allowed to gently wash through the chamber and exchange for the ethanol in the tissue. When the exchange was complete, the $CO_2$ was brought to its critical point of 1073 psi and 31 °C and allowed to gently bleed away. The dry sample pieces were secured, fracture side up, to labeled SEM stubs. The stubs were sputter-coated with 15 nm chromium using a Denton DV-602 magnetron sputter coater. The samples were imaged at 10 kV using the upper stage of a Topcon DS130 field emission scanning electron microscope and images were collected using a Quartz PCI digital image collection system.

**Hepatic artery and portal vein blood flow analysis by ultrasound Doppler imaging.** The animals were maintained on a 1.5–2% isofluorane anesthesia and positioned on the platform in the supine position. Respiratory physiology, electrocardiogram, and body temperature was monitored during the procedure. Hepatic artery and portal vein blood flow hemodynamics were analyzed using a Vevo 2100 high frequency, digital, linear array, color Doppler, small animal ultrasound machine using the B-mode, pulsed wave, and color Doppler mode. The movement of blood was reflected in a change in the pitch of the reflected sound waves (called the Doppler Effect). The signals generated were displayed in graphs or color pictures. Color Doppler mode allowed for rapid identification of arteries and veins with ease and enabled accurate sampling and quantification of the blood flow. Peak systolic value (PSV), low diastolic value (LDV), and mean velocity (MV) were used to calculate the RI and PI. The following formulae were used: RI = PSV–LDV/PSV; pulsatility index (PI) = PI = PSV–LDV/MV. The maximum velocity of the portal vein was also measured. The Doppler gate length was 5 mm and the angle between the Doppler ultrasound beam and the long axis of the

hepatic portal vein was maintained between 300 and 600. The spectral waveform of the hepatic portal vein was recorded for at least 5 s using an MS 550D transducer for abdominal imaging.

**Statistics and reproducibility**. Statistical analyses were carried out using the GraphPad Prism 6.0 software. The number of animals is specified in each figure and legend. All in vitro experiments were carried out five times minimum. For image quantifications and other analyses, statistical significance was assayed by either Student's $t$ test and/or one-way analysis of variance for multiple comparisons followed by post hoc Tukey's test. Box plots show range, median, and quartiles. In all figures, $*P < 0.05$; $**P < 0.01$; $***P < 0.001$; $****P < 0.0001$; ns. denotes not significant.

**Reporting summary**. Further information on research design is available in the Nature Research Reporting Summary linked to this article.

## Data availability
Source data are available with this paper as Supplementary data 1.

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

## Acknowledgements
We thank Yinwei Zhang, David Brenner, and Liangwei Li for excellent suggestions for our studies. We thank Jeanette Taylor for her assistance in SEM imaging. This work is supported in part by research grants from the National Institute of Health (CA175112, CA118113, CA178730) and Georgia Cancer Coalition to Z.-R.L., R.C.T. and M.S. are supported by an MBD fellowship, GSU.

## Author contributions
Z.-R.L. conceptualized, planned, and coordinated the study. Z.-R.L. wrote the paper. R.C.T. conceptualized, planned, conducted most of the experiments, data analyses, and participated in paper writing; G.S. and M.S. participated in liver fibrosis induction, animal treatment experiments, tissue section staining, and data analyses. J.Y. participated in collagen analyses of fibrotic livers. W.S, C.L. and L.S. helped in protein expression and

purification. H.Y. and H.E.G. performed part of IF and H&E staining and histology analyses. A.B.F. helped in liver pathological analyses and collagen staining and data analyses. J.G-S. critically reviewed the manuscript and conceived experiments. All authors discussed the results and commented on the paper.

## Competing interests

Z.-R.L. holds shares in the company ProDa BioTech LLC, which licensed the rights to commercialize ProAgio. L.S. holds shares in the company Amoytop Biotech Co. Ltd. which licensed the rights to commercialize ProAgio in China. The remaining authors declare no competing financial and personal interests.
