## [Peer Review File · Communications Biology]

Reviewers' comments:

Reviewer #1 (Remarks to the Author):

In this manuscript Turaga and colleagues report a strategy for CLD treatment by induction of integrin avb3 mediated cell apoptosis using a rationally designed protein (ProAgio). The authors rightly highlight that there is a need to develop new therapeutics to treat CLD and provide convincing evidence that ProAgio can reverse liver fibrosis and relieve blood flow resistance in mouse models. The paper is well structured and there is a clear narrative. There however is not sufficient evidence to confirm ProAgio works by specifically targeting activated HSC and capillarized LSECs for avb3 mediated cell apoptosis. Given that this is an important part of the manuscript further work is required to convince the reader of the mechanism. Furthermore, if this was to be considered as a future therapeutic to treat human CLD the manuscript would benefit from more evidence that avb3 integrin is expressed on the appropriate populations in human disease.

Comments

1. The upregulation of integrin av and Integrin b3 in activated human HSC in vitro is convincing. However there needs to be more work on characterising the levels of integrin av and integrin b3 in healthy and cirrhotic human liver, and healthy and fibrotic mouse liver to convince the reader that (a) avb3 is upregulated in activated HSC and capillarized LSECs in vivo, and (b) it is specific to these cell types. Topographical identification of Integrin av and Integrin b3 would help answer this question. This has previously been looked at in mouse and rat (Dobie 2019, Cell Reports; Wu, Jain 2011, Hepatology). The former study also contains expression data that suggests both Itgav and Itgb3 are not elevated in mouse activated HSC. Online web browsers of human healthy and cirrhotic liver could also be used to further interrogation of expression patterns of ITGAV and ITGB3 integrin across different lineages (Aizarani 2019, Nature; Ramachandran 2019, Nature).
2. Despite this work following on from a previous paper developing ProAgio there needs to be more evidence or explanation as to the specificity of ProAgio. Previous reports show that av integrin has five possible binding partners (b1, b3, b5, b6 and b8) with avb1, avb3, avb5, and avb8 expressed on HSC (Henderson 2014, Nature Medicine). The authors need to show that ProAgio is specific to avb3 and does not interact with the other av integrins at the concentrations used. This is particularly important as previous reports indicate that ProAgio may also weakly interact with avb5.
3. The authors state that ProAgio treatment leads to apoptosis of aHSC and caLSEC in fibrotic liver. The staining could however be more convincing. The authors should take tissue at an earlier timepoint in treatment and perform co-staining of aSMA or CD31 and cleaved caspase-3 to get a clearer indication if ProAgio is causing increased apoptosis of these populations. This should also be compared to vehicle treated animals.
4. The staining of CD31 in Figure 5 is surprising. There are number of papers that show strong CD31 staining of LSECs in healthy and fibrotic mouse liver (e.g. Bonomini 2013, Biogerontology; Mederacke 2013, Nature Communications). This is not observed here. The differences observed make the results difficult to interpret and should be explained.
5. According to the data integrin av and b3 are detectable in qHSC, but when ProAgio is added to culture media of these cells no apoptosis is observed. Can the authors please clarify why they see no apoptosis in qHSC despite the relevant integrins being present.
6. The final section of the manuscript results in an important one and should be expanded on and possibly incorporated earlier in the manuscript. It would be nice to see confirmation that av integrin and b3 are not present on all other cell lineages (hepatocytes, cholangiocytes, Kupffer cells, LSECs, quiescent HSC), and that ProAgio does not affect apoptosis in these populations in healthy and disease. The authors should also include histological analysis on HSC (quiescent) to assess the status of HSC in response to vehicle and ProAgio treated healthy animals.

7. The authors nicely plot the majority of the graphs, so each individual data point is visible which highlights just how sticking some of the data is. It would be helpful if the statistical test used and what each asterix represents (i.e * = $p < 0.05$) was presented in each figure legend also. Can the authors also confirm in the text that the data was checked for normal distribution before deciding on a parametric or non parametric followup test.

Reviewer #2 (Remarks to the Author):

In the submitted manuscript the authors show that the protein 'ProAgio', designed to bind integrin $\alpha v \beta 3$ and induce cell apoptosis was effective at promoting apoptosis of activated HSCs reversing fibrosis. These data are interesting, and important to the treatment of chronic liver disease, and would be of interest to the field. The impact of the manuscript could be improved by addressing the following items.

1. The authors show that ProAgio has no impact on primary hepatocyte or kupffer cell viability; however, ProAgio also had limited effects on viability of quiescent HSCs. As the authors propose ProAgio as a treatment for chronic liver disease, hepatocyte apoptosis should be assessed in vivo.
2. The CCl4 and TAA models are excellent models of hepatotoxic liver fibrosis, but are not pathologically similar to chronic liver disease. The inclusion of a metabolic-type liver disease model or cholestatic liver disease model would broaden the impact of the manuscript.
3. Fibrosis regression is observed with removal of the injurious stimulus in vivo, and early stages of fibrosis regression involve apoptosis of activated HSCs. While TAA-induced fibrosis is more persistent than CCl4-fibrosis, the authors should include the vehicle treated group in Figures 3H and 5E to compare HSC apoptosis and sinusoidal cell apoptosis in vehicle and ProAgio treated groups after removal of the TAA/EtOH.
4. Appropriate controls for the Co-IP should be included, including input and IgG.
5. The authors should review the manuscript to ensure all necessary information is present, including, but not limited to, where primary HSCs were purchased from.

We thank reviewer for reviewing our manuscript. We finally finished all experiments and revised our manuscript to address reviewers' concerns. Please see below our point-by-point responses to reviewer's critiques. The changes are in italic text in the manuscript file and supplementary.

Reviewer 1

1. The upregulation of integrin αv and Integrin $\beta 3$ in activated human HSC in vitro is convincing. However there needs to be more work on characterising the levels of integrin αv and integrin $\beta 3$ in healthy and cirrhotic human liver, and healthy and fibrotic mouse liver to convince the reader that (a) $\alpha v\beta 3$ is upregulated in activated HSC and capillarized LSECs in vivo, and (b) it is specific to these cell types. Topographical identification of Integrin αv and Integrin $\beta 3$ would help answer this question. This has previously been looked at in mouse and rat (Dobie 2019, Cell Reports; Wu, Jain 2011, Hepatology). The former study also contains expression data that suggests both Itgav and Itgb3 are not elevated in mouse activated HSC. Online web browsers of human healthy and cirrhotic liver could also be used to further interrogation of expression patterns of ITGAV and ITGB3 integrin across different lineages (Aizarani 2019, Nature; Ramachandran 2019, Nature).

Response: *We first analyzed the integrin $\beta 3$ expression levels in different cell types isolated from normal healthy or cirrhotic livers of murine TAA model by RT-PCR. The integrin was not expressed in HSC, LSEC, and hepatocyte in normal healthy liver. The integrin was expressed in Kupffer cells of normal healthy liver. However, in fibrotic liver, the integrin was expressed in HSC, LSEC, and Kupffer cells but not in hepatocytes (see Fig. S1A in revision). We further re-analyzed the scRNA-seq data from public domain (from Ramachandran et.al.) and validated the expression of integrin $\alpha V\beta 3$ in multi-lineage approach in various cell types involved in liver physiology and pathology (<https://www.nature.com/articles/s41586-019-1631-3>). Our analyses indicated that, although integrin αv was expressed in a number of cell lineages of both normal and fibrotic liver, integrin $\beta 3$ was not expressed or expressed in low levels in most cells in liver (please see Fig S1 B&C in revision).*

2. Despite this work following on from a previous paper developing ProAgio there needs to be more evidence or explanation as to the specificity of ProAgio. Previous reports show that αv integrin has five possible binding partners ($\beta 1$, $\beta 3$, $\beta 5$, $\beta 6$ and $\beta 8$) with $\alpha v\beta 1$, $\alpha v\beta 3$, $\alpha v\beta 5$, and $\alpha v\beta 8$ expressed on HSC (Henderson 2014, Nature Medicine). The authors need to show that ProAgio is specific to $\alpha v\beta 3$ and does not interact with the other αv integrins at the concentrations used. This is particularly important as previous reports indicate that ProAgio may also weakly interact with $\alpha v\beta 5$.

Response: *Our previous report (Nature Communications 7, 11675 (2016)) tested the interaction of ProAgio with integrin pairs, $\alpha 1\beta 1$, $\alpha 11\beta 3$, $\alpha v\beta 3$, $\alpha v\beta 5$, $\alpha v\beta 6$ by either ELISA/SPR binding or cell attachment assays (please see Figure 1 and Figure S2 in our published paper). We further carried out Co-IP of ProAgio with $\beta 3$, $\beta 5$, $\beta 1$, and $\beta 6$ (see Figure S2F in revision). Our results demonstrated that ProAgio only co-IP with $\beta 3$, confirmed our*

conclusion that ProAgio specifically interacts with integrin $\alpha\beta3$.

3. The authors state that ProAgio treatment leads to apoptosis of aHSC and caLSEC in fibrotic liver. The staining could however be more convincing. The authors should take tissue at an earlier timepoint in treatment and perform co-staining of aSMA or CD31 and cleaved caspase-3 to get a clearer indication if ProAgio is causing increased apoptosis of these populations. This should also be compared to vehicle treated animals.

Response: *We carried out the analyses suggested by the reviewer (please see Fig. 3H and Fig. 5E in revision). The results indicated that CC3 co-stains with aSMA and CD31 in fibrotic livers of ProAgio groups but not in the vehicle treated groups.*

4. The staining of CD31 in Figure 5 is surprising. There are number of papers that show strong CD31 staining of LSECs in healthy and fibrotic mouse liver (e.g. Bonomini 2013, Biogerontology; Mederacke 2013, Nature Communications). This is not observed here. The differences observed make the results difficult to interpret and should be explained.

Response: *To address the reviewers' comment, we re-done the IHC staining. The results are consistent (please see Fig. 5C). It is a debatable issue whether CD31 (PECAM-1) is expressed in LSEC in normal healthy liver. Our results are consistent with several reports that CD31 (PECAM-1) expression is negative or low in LSEC of normal healthy liver (Johanne Poisson, et.al. Journal of Hepatology Vol 66 (1) pp 212-227 (2017); Couvelard, A. et.al. Am J Pathol. 1993 Sep; 143(3): pp738–752.). The inconsistent observations may resulted from using different antibodies and analyzing methods in the various experiments. This explanation has been added in discussion section.*

5. According to the data integrin αv and $\beta 3$ are detectable in qHSC, but when ProAgio is added to culture media of these cells no apoptosis is observed. Can the authors please clarify why they see no apoptosis in qHSC despite the relevant integrins being present.

Response: *No or very low levels of integrin $\beta 3$ is expressed in qHSC. Integrin $\alpha\beta 3$ expression in activated HSC is strong increased (this is consistent with a number of previous reports, Patsenker, E., et al., Hepatology, 2009. 50(5): p. 1501-11; Zhou, X., et al., J Biol Chem, 2004. 279(23): p. 23996-4006; Li, F., Hepatology, 2011. 54(3): p. 1020-30). ProAgio only induces apoptosis of the cells with high levels of integrin $\alpha v\beta 3$ expression (please see our previous paper in Nature Communications, 7, 11675 (2016)).*

6. The final section of the manuscript results in an important one and should be expanded on and possibly incorporated earlier in the manuscript. It would be nice to see confirmation that αv integrin and $\beta 3$ are not present on all other cell lineages (hepatocytes, cholangiocytes, Kupffer cells, LSECs, quiescent HSC), and that ProAgio does not affect apoptosis in these populations in healthy and disease The authors should also include histological analysis on HSC (quiescent) to assess the status of HSC in response to vehicle and ProAgio treated healthy animals.

Response: *We carried out RT-PCR to probe integrin $\beta 3$ expression in different cell types, including HSC, LSEC, hepatocytes, and Kupffer cells, in normal healthy liver and fibrotic liver (see Fig. S1A in revision). Clearly, our results showed that the integrin is not expressed in HSC,*

LSEC, and hepatocyte in normal healthy liver. The integrin is expressed in HSC, LSEC, and Kupffer cells in fibrotic liver. We further re-analyzed the scRNA-seq data from public domain (from Ramachandran et.al.) and validated the expression of integrin $\alpha V\beta 3$ in multi-lineage approach in various cell types involved in liver physiology and pathology (<https://www.nature.com/articles/s41586-019-1631-3>). Our analyses indicated that, although integrin αv was expressed in a number of cell lineages of both normal and fibrotic liver, integrin $\beta 3$ was not expressed or expressed in low levels in most cells in liver (Fig S1C in revision).

7. The authors nicely plot the majority of the graphs, so each individual data point is visible which highlights just how sticking some of the data is. It would be helpful if the statistical test used and what each asterisk represents (i.e * = $p < 0.05$) was presented in each figure legend also. Can the authors also confirm in the text that the data was checked for normal distribution before deciding on a parametric or non parametric followup test.

Response: We added statistical meaning of each asterisk in each figure legend. We have updated the statistical methods in methods section.

Reviewer 2

1. The authors show that ProAgio has no impact on primary hepatocyte or kupffer cell viability; however, ProAgio also had limited effects on viability of quiescent HSCs. As the authors propose ProAgio as a treatment for chronic liver disease, hepatocyte apoptosis should be assessed in vivo.

Response: We performed IHC staining of cleaved caspase-3 in sections from the vehicle and ProAgio (20 mg/kg) treated normal mice (without fibrosis induction), and no positive stain was observed (see Fig. S6G).

2. The CCl₄ and TAA models are excellent models of hepatotoxic liver fibrosis, but are not pathologically similar to chronic liver disease. The inclusion of a metabolic-type liver disease model or cholestatic liver disease model would broaden the impact of the manuscript.

Response: We agree with the reviewer. However, no single animal model recapitulates all pathological features of human chronic liver diseases. CCl₄ and TAA models are very commonly used models for liver fibrosis and cirrhosis. It may be difficult to test an agent with all different models. We thank the reviewers' valuable suggestion. We are pursuing a more extensive follow-up study with different models, particularly with metabolic-type liver disease model and cholestatic liver disease model as suggested by the reviewer.

3. Fibrosis regression is observed with removal of the injurious stimulus in vivo, and early stages of fibrosis regression involve apoptosis of activated HSCs. While TAA-induced fibrosis is more persistent than CCl₄-fibrosis, the authors should include the vehicle treated group in Figures 3H and 5E to compare HSC apoptosis and sinusoidal cell apoptosis in vehicle and ProAgio treated groups after removal of the TAA/EtOH.

Response: *We included the fluorescence images of the vehicle treated group to compare to ProAgio and represent HSC and sinusoidal cell apoptosis (please see Fig. 3H and Fig. 5E in revision).*

4. Appropriate controls for the Co-IP should be included, including input and IgG.

Response: *New Co-IP experiments were performed with IgG as controls (please see Figure S2E).*

5. The authors should review the manuscript to ensure all necessary information is present, including, but not limited to, where primary HSCs were purchased from.

Response: *We have carefully checked our manuscript. All necessary information is present in the manuscript. Thank you.*

We hope that our point-by-point responses and new experimental results satisfactorily address all reviewer's critiques.

Sincerely,

Dr. Zhi-Ren Liu, Ph.D

Professor of Cell Biology

REVIEWERS' COMMENTS:

Reviewer #1 (Remarks to the Author):

The authors have addressed the comments I made.

Reviewer #2 (Remarks to the Author):

The authors have addressed all concerns. The manuscript should be of interest to a broad audience.